# MusicDET: Zero-Shot AI-Generated Music Detection

**Chaolei Han** [1]   **Hongsong Wang** [2][3]   **Jie Gui** [1][4][5]

## Abstract

Detecting AI-generated music is crucial for preserving artistic authenticity and preventing the misuse of generative music technologies. However, existing discriminative detectors typically rely on generated samples during training and often suffer from severe performance degradation when confronted with music produced by unseen generators, which limits their real-world applicability. To address this issue, we formulate a zero-shot setting for AI-generated music detection, where the detector is trained exclusively on real music without access to any generated samples. Under this setting, we propose MusicDET, a generator-agnostic detection framework based on frequency-guided normalizing flows that probabilistically models the distribution of real music features. By evaluating the likelihood of an input sample under the learned real-music distribution, MusicDET enables effective detection of out-of-distribution music signals. Experiments on the FakeMusicCaps and SONICS datasets show that MusicDET consistently outperforms conventional discriminative detectors, particularly when detecting music generated by previously unseen models. The code is at https://github.com/Chaolei98/MusicDET.

## 1. Introduction

With the rapid progress of generative artificial intelligence, AI-generated music increasingly permeates music creation,

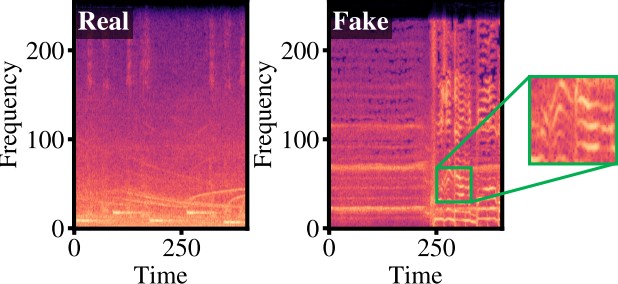

*Figure 1.* **Systematic discrepancies between real music and AI-generated music in terms of energy spectrograms.** Real music exhibits coherent and well-organized time–frequency structures, whereas AI-generated music often displays irregular and less consistent spectral energy patterns.

distribution, and consumption (Zhang et al., 2025; Schneider et al., 2024; Tian et al., 2026; Bryan-Kinns et al., 2024). While these technologies significantly improve the efficiency and diversity of music production, they also introduce growing risks of misuse and copyright disputes, including blurred authorship, value erosion, and cultural homogenization. These challenges threaten the fairness of the music ecosystem and highlight the need for methods to distinguish human-created music from AI-generated content.

As a passive forensic technology, AIGM detection aligns well with the present landscape of inadequate market regulation. Although audio deepfake detection (ADD) (Li et al., 2024a; Zhang et al., 2024; Zhu et al., 2024; Zhang et al., 2023) develops strong benchmarks and methods for speech authenticity, these approaches typically rely on speech-specific low-level cues learned from voice conversion and text-to-speech artifacts. Music, in contrast, is organized around richer structures and expressions, including melody, harmony, rhythm, timbre, and lyrics, and it varies widely across genres and production styles (Li et al., 2024b). As a result, directly applying speech-oriented detectors to music often yields suboptimal performance, which calls for detection methods tailored to AI-generated music.

Despite its importance, AI-generated music detection remains underexplored. Existing methods (Rahman et al., 2025; Afchar et al., 2025; Xie et al., 2026; Kim & Go, 2025) are primarily engineered to capture artifacts specific to particular generators, achieving high accuracy in closed-set

---

[1]School of Cyber Science and Engineering, Southeast University, Nanjing 210096, China [2]School of Computer Science and Engineering, Southeast University, Nanjing 210096, China [3]Key Laboratory of New Generation Artificial Intelligence Technology and Its Interdisciplinary Applications (Southeast University), Ministry of Education, China [4]Purple Mountain Laboratories, Nanjing 210000, China [5]Engineering Research Center of Blockchain Application, Supervision And Management (Southeast University), Ministry of Education, China. Correspondence to: Hongsong Wang <hongsongwang@seu.edu.cn>, Jie Gui <guijie@seu.edu.cn>.

*Proceedings of the 43rd International Conference on Machine Learning*, Seoul, South Korea. PMLR 306, 2026. Copyright 2026 by the author(s).

scenarios. However, their performance drops sharply when evaluated across unseen generators. Evaluation under open-set scenarios is therefore essential, as generative models continue to proliferate and developing a separate detector for each generator is impractical. *This challenge naturally motivates a zero-shot AIGM detection setting, where detectors are required to identify AI-generated music without relying on any generator-specific training data.*

Human experts offer an instructive analogy. Musicians often identify AI-generated music more accurately than non-expert listeners because they have extensive exposure to real music and a refined sense of authentic timbre and musical organization. Inspired by this observation, we aim to build a detector that leverages prior knowledge of real music to assess the authenticity of previously unseen pieces. Normalizing flows (Zhai et al., 2025) provide a natural foundation for this goal. They model the probability distribution of real samples via invertible transformations and quantify departures from normality through likelihood, which makes them attractive for detecting forged content. While normalizing flows are widely used in anomaly detection (Hirschorn & Avidan, 2023; Dai & Chen, 2022; Chiu & Lai, 2023), applying them to music introduces distinctive challenges because real music spans diverse genres, instrumentation, and production styles, resulting in a complex and highly multimodal distribution.

A key requirement is therefore a representation that preserves perceptually salient musical attributes and yields a distribution that is amenable to likelihood-based modeling. The time–frequency energy spectrum meets this need because it encodes regularities that are closely tied to musical perception, including harmonic organization, rhythmic periodicity, and timbral texture. As shown in Figure 1, real music and AI-generated music exhibit systematic differences in energy spectra. Real music tends to present coherent and well-organized time–frequency structures, whereas AI-generated music more often shows irregular or less consistent spectral energy patterns. These properties make energy-based representations particularly suitable for likelihood modeling because they help characterize the structure of real music while highlighting deviations introduced by generative processes. Motivated by these observations, **we propose MusicDET, a zero-shot music detector based on normalizing flows.** MusicDET learns an invertible mapping that projects real music energy-spectral features into a simple prior distribution using only real music during training, thereby enabling effective detection of generated music as deviations from the learned distribution. MusicDET can also be extended to a supervised, class-conditional formulation that learns class-aware distributions for real and AI-generated music by maximizing their respective likelihoods, which further improves discriminability when labeled generated samples are available. Experiments on FakeMusicCaps

(Comanducci et al., 2025) and SONICS (Rahman et al., 2025) show that MusicDET achieves state-of-the-art performance under cross-generator evaluation protocols. Additional evaluations on ASVSpoof2019LA (Todisco et al., 2019) and CtrSVDD (Zang et al., 2024) suggest that the proposed approach transfers beyond music detection and is effective for broader audio authenticity and anomaly-related tasks. Our contributions can be summarized as follows:

- **A More practical problem:** We introduce zero-shot AI-generated music detection, which only uses real samples for training to enhance generalization.

- **Novel framework for AI-generated music detection:** We propose frequency-guided normalizing flows, a likelihood-based framework that models real-music distributions in the time–frequency energy domain and enables robust, generator-agnostic detection of AI-generated music.

- **State-of-the-art open-set performance:** Extensive experiments demonstrate that MusicDET consistently achieves state-of-the-art results under cross-generator evaluation protocols.

**Conflict of Interest Disclosure.** The authors declare no financial conflicts of interest related to this work.

## 2. Related Work

### 2.1. AI-Generated Music Detection

Existing research on fake music predominantly focuses on developing music generation models rather than detection (Schneider et al., 2024; Qiang et al., 2026). For instance, MusicGen (Copet et al., 2023b) adopts a single-stage transformer architecture to generate high-fidelity audio directly from textual or melodic prompts, enabling efficient end-to-end text-to-music generation without cascaded vocoding. MusicLM (Agostinelli et al., 2023) extends this paradigm through a hierarchical audio-token modeling pipeline that aligns semantic text embeddings with long-duration musical structure, and it achieves strong temporal coherence and sound quality. Building on diffusion techniques, MSDM (Xu et al., 2024) conditions the generation process on multiple heterogeneous modalities, which allows fine-grained control over rhythm, emotion, and style. Similarly, JASCO (Tal et al., 2024) integrates symbolic and audio embeddings within a diffusion framework to improve temporal synchronization and structural consistency. MusiCoGen (Lan et al., 2024) introduces explicit rhythm and chord conditioning in a transformer-based model, enabling controllable generation aligned with musical meter and harmonic progression.

Compared with advances in generative algorithms, research on AI-generated music detection remains relatively underde-

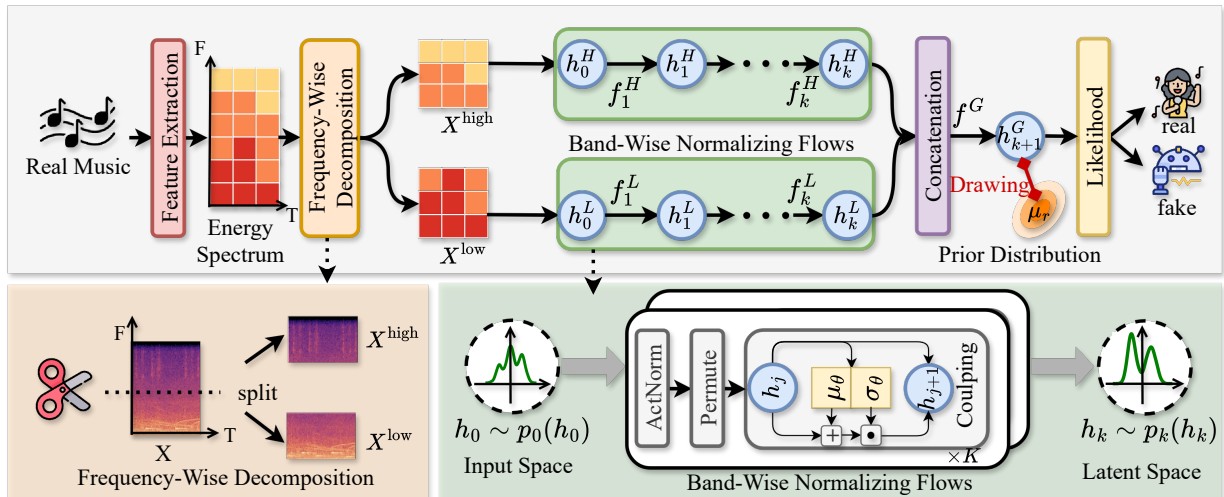

*Figure 2.* **Pipeline of MusicDET.** In the zero-shot setting, MusicDET converts real-music waveforms into energy spectrograms and decomposes the resulting features into multiple frequency sub-bands via Frequency-Wise Decomposition. Each sub-band is processed by an independent Band-Wise Normalizing Flow, which performs invertible transformations to learn band-specific representations. The resulting latent codes are then concatenated and fed into a Global Normalizing Flow to model their joint distribution under a predefined prior. At inference time, MusicDET classifies a query by computing its log-likelihood under the learned flow model, where low-likelihood samples are flagged as AI-generated music.

veloped, and existing studies primarily emphasize improving detection accuracy. Xie et al. (Xie et al., 2026) propose a wavelet prompt tuning method that aims to capture type-invariant auditory deepfake cues from diverse audio inputs. However, its performance on music tasks remains limited due to the lack of task-specific design. Rahman et al. (Rahman et al., 2025) generate end-to-end synthetic music using the Suno and Udio models and introduce a detection framework that models long-range temporal dependencies. Kim et al. (Kim & Go, 2025) segment long music recordings into shorter clips and learn inter-segment relationships, achieving high classification accuracy. Afchar et al. (Afchar et al., 2025) use the FMA dataset together with multiple autoencoder-based generators to synthesize music, and they show that current detection approaches fail to generalize effectively across different generative models. Although the aforementioned discriminative detection models achieve high accuracy for forgeries produced by a specific generator, their performance degrades markedly when evaluated across generators. This work addresses the challenge of detecting AI-generated music under cross-generator conditions.

## 2.2. Generalizable Audio Deepfake Detection

Audio deepfake detection (Wang et al., 2025b; Li et al., 2024c) increasingly emphasizes generalization to unseen conditions. Xie et al. (Xie et al., 2026) align real speech across domains and separate fake speech using adversarial learning and triplet loss to improve out-of-distribution (OOD) performance. Zhu et al. (Zhu et al., 2024) lever-

age style–linguistics mismatch learned from real-only pretraining, achieving more generalizable and interpretable detection. Yang et al. (Yang et al., 2024) show that large-scale pretrained representations generalize better than hand-crafted features, and they further enhance robustness via multi-view feature fusion. Wang et al. (Wang et al., 2025a) fuse Wav2vec and WavLM with cross-attention and introduce wavelet-style token mixing to strengthen cross-dataset detection. In contrast to these speech-focused studies, we formulate a stricter zero-shot setting for AI-generated music detection that targets unseen generators and diverse musical structures without domain-specific adaptation.

## 2.3. Normalizing Flows

Normalizing flows are a class of generative models that transform complex real-world data distributions into simple, typically Gaussian, distributions through a sequence of invertible and differentiable mappings. Rudolph et al. (Rudolph et al., 2021) are among the first to apply normalizing flows to anomaly detection by modeling the distribution of pretrained features. Hirschorn et al. (Hirschorn & Avidan, 2023) apply normalizing flows to human pose graph sequences for anomaly detection, and Yao et al. (Yao et al., 2024) propose a hierarchical Gaussian mixture framework to unify anomaly detection. Despite these advances in computer vision, the use of normalizing flows for detection in the audio domain remains largely unexplored. To the best of our knowledge, this work is the first to leverage normalizing flows for AI-generated music detection.

# 3. Method

## 3.1. Zero-Shot AI-Generated Music Detection

We define zero-shot AI-generated music detection as the task of distinguishing real music from AI-generated music when the detector is trained only on real music. Let $\mathcal{D}_{\text{real}}$ denote the distribution of real music. The training set samples $x \sim \mathcal{D}_{\text{real}}$ and includes no AI-generated examples. At test time, the detector takes a music sample $x$ and outputs a score $s(x)$ indicating whether $x$ is AI-generated. Evaluation is conducted under cross-generator conditions, where AI-generated test samples come from one or more generators unseen during training and development. This setting emphasizes real-only training and generator-agnostic generalization as generative models evolve.

## 3.2. MusicDET

As illustrated in Figure 2, we propose **MusicDET**, a one-class, flow-based framework for zero-shot AI-generated music detection. The model is trained exclusively on real music and identifies AI-generated music as samples that deviate from the learned data distribution.

**Feature Extraction:** Given a raw music waveform segment, we first resample it to a unified sampling rate and crop or pad it to a fixed length, yielding $x_{\text{wav}} \in \mathbb{R}^L$. We then compute the short-time Fourier transform (STFT) and obtain the corresponding power spectrum. To preserve the time–frequency structure that is fundamental to musical signals, including harmonic organization, rhythmic regularity, and timbral texture, we apply convolutional layers to extract spectral energy features $X \in \mathbb{R}^{B \times C \times T \times F}$, where $B$ denotes the batch size, $C$ the number of channels, $T$ the number of time frames, and $F$ the number of frequency bins. Each instance in the batch is treated as a sample $x \in \mathcal{X}$, where $\mathcal{X}$ denotes the input data space.

**Frequency-Wise Decomposition:** Music exhibits heterogeneous statistical characteristics across frequency bands. Low-frequency components are closely related to rhythm and fundamental pitch, whereas high-frequency components encode timbre, overtones, and transient details. Modeling the entire spectrum with a single flow often leads to unstable density estimation due to such mixed statistics.

To explicitly account for this structure, we decompose the spectral feature along the frequency axis into multiple bands. For simplicity, we illustrate the idea using two bands as an example:

$$X = [X^{\text{low}}, X^{\text{high}}], \qquad (1)$$

where $X^{\text{low}} \in \mathbb{R}^{B \times C \times T \times F_{\text{low}}}$, $X^{\text{high}} \in \mathbb{R}^{B \times C \times T \times F_{\text{high}}}$, and $F_{\text{low}} + F_{\text{high}} = F$.

This decomposition does not impose independence assumptions between bands. Instead, it reorganizes the input space into frequency-localized subspaces that facilitate more expressive and stable likelihood modeling.

**Band-Wise Normalizing Flows:** Each frequency band is processed by an independent normalizing flow:

$$f_\theta^{\text{low}} : x^{\text{low}} \leftrightarrow h_K^{\text{low}}, \qquad f_\theta^{\text{high}} : x^{\text{high}} \leftrightarrow h_K^{\text{high}}. \qquad (2)$$

Each flow follows a Glow-style architecture (Kingma & Dhariwal, 2018), consisting of $K$ flow steps. Each step includes ActNorm, an invertible $1 \times 1$ convolution, and an affine coupling layer. The band-wise flows act as invertible reparameterizations that reorganize spectral energy patterns within each frequency range, capturing music-specific regularities such as smooth harmonic evolution in low frequencies and fine-grained timbral variations in high frequencies.

**Global Flow:** While band-wise flows model local spectral characteristics, musical coherence arises from strong cross-frequency dependencies, such as the alignment between fundamental frequencies and their harmonics. To capture such global structure, we concatenate the band-wise latent representations:

$$h_K = \text{Concat}(h_K^{\text{low}}, h_K^{\text{high}}), \qquad (3)$$

and feed the resulting representation into a global normalizing flow $f_\theta^{\text{global}}$. The global flow defines the final bijection between input space $\mathcal{X}$ and latent space $\mathcal{Z}$ and is trained with a Gaussian prior

$$p_Z(z) = \mathcal{N}(\mu_{\text{real}}, I), \qquad (4)$$

where $\mu_{\text{real}}$ denotes the latent center of real music.

**Likelihood Calculation:** Then, the data likelihood can be computed exactly using the change-of-variables formula (Villani, 2003):

$$p_X(x) = p_Z\big(f_\theta(x)\big) \left| \det J_{f_\theta}(x) \right|, \qquad (5)$$

where $f_\theta$ denotes the overall transformation and $J_{f_\theta}(x)$ denotes the Jacobian matrix of $f_\theta$ with respect to $x$. Taking logarithms yields

$$\log p_X(x) = \log p_Z(h_{K+1}) + \sum_{j=1}^{K+1} \log \left| \det J_{f_j}(h_{j-1}) \right|, \qquad (6)$$

where the intermediate states $h_0 = x$ and $h_j = f_j(h_{j-1})$ for $j = 1, \ldots, K+1$. To ensure computational efficiency, each transformation $f_j$ is designed such that its Jacobian determinant can be evaluated tractably, as in coupling-based flow architectures.

**Training and Inference:** The entire framework is trained by minimizing the negative log-likelihood (NLL) of real

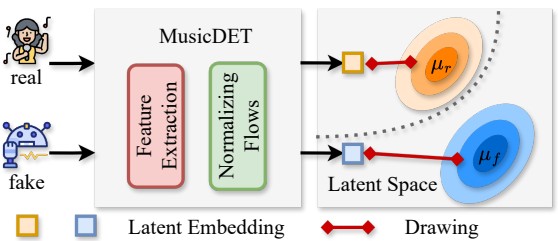

*Figure 3.* **Pipeline of class-conditional MusicDET.** The normalizing flow learns class-conditional probability density functions for real music and AI-generated music via invertible transformations, enabling detection through likelihood estimation.

music samples:

$$\min_\theta \ \mathbb{E}_{x \sim \mathcal{D}_{\text{real}}}\left[-\log p_X(x)\right]. \quad (7)$$

At inference time, samples with low likelihood under the learned distribution are identified as AI-generated music.

### 3.3. Class-Conditional MusicDET

To further enhance discriminative capability, we extend MusicDET to a class-conditional density modeling framework, as illustrated in Figure 3. This extension removes the training-time constraint of not having access to AI-generated music samples. Importantly, MusicDET remains a detector rather than a classifier: the flow transformations are shared across classes, and class information is introduced solely through the latent prior. Let $y \in \{\text{real}, \text{fake}\}$ denote the class label. After obtaining the concatenated latent representation $h_K$, we apply the same global flow $f_\theta^{\text{global}}$ and assume a class-dependent Gaussian prior

$$p_{Z|Y}(z \mid y) = \mathcal{N}(\mu_y, I), \quad (8)$$

where $\mu_{\text{real}}$ and $\mu_{\text{fake}}$ are the latent means for real music and AI-generated music, respectively. The conditional log-likelihood is given by

$$\log p_{X|Y}(x \mid y) = \log p_{Z|Y}(h_{K+1} \mid y)$$
$$+ \sum_{j=1}^{K+1} \log\left|\det J_{f_j}(h_{j-1})\right|. \quad (9)$$

During training, both real music and AI-generated music samples are used to estimate the class-specific latent means $\mu_{\text{real}}$ and $\mu_{\text{fake}}$. The flow parameters $\theta$ are optimized by minimizing the negative conditional log-likelihood over all training samples:

$$\min_\theta \ \mathbb{E}_{(x,y) \sim \mathcal{D}_{\text{train}}}\left[-\log p_{X|Y}(x \mid y)\right]. \quad (10)$$

Despite being trained with class-conditional information, MusicDET operates as a detector at inference time. Given a test sample with unknown label, we evaluate its likelihood under the real music prior only:

$$\log p_X(x \mid y = \text{real}). \quad (11)$$

AI-generated music, although observed during training, typically occupies latent regions that deviate from the real-music center $\mu_{\text{real}}$ and therefore receives lower likelihood values under the real-music prior.

## 4. Experiment

### 4.1. Datasets and Experimental Setup

**Dataset:** FakeMusicCaps (Comanducci et al., 2025) is a benchmark dataset for detecting and attributing text-to-music (TTM) generated audio. It uses approximately 5.5k expert-written captions from MusicCaps (Agostinelli et al., 2023) as prompts and synthesizes one 10-second clip per caption with five TTM generators (Copet et al., 2023a; Chen et al., 2024a; Liu et al., 2024; Evans et al., 2025; Melechovsky et al., 2024). This procedure produces 27,605 generated tracks in total. SONICS (Rahman et al., 2025) is a large-scale dataset for end-to-end detection of AI-generated music that spans diverse musical and lyrical styles. Its real-music subset is constructed by randomly sampling song metadata from the Genius Lyrics Dataset[1] and downloading the corresponding tracks from YouTube, resulting in 48,090 songs from 9,096 artists. Its AI-generated subset is synthesized using two major families of commercial generation systems, Suno[2] and Udio[3], including Suno v2, v3, and v3.5, as well as Udio 32 and Udio 130. We follow the train/validation/test split protocol used in prior work (Xie et al., 2026; Rahman et al., 2025). Based on this split, we further separate the AI-generated samples in each subset by generator, creating one generator-specific fake partition per subset, while keeping the real samples unchanged and shared across generators. This design supports cross-generator evaluation. For ASVspoof 2019 LA (Todisco et al., 2019), the training and development sets include spoofing systems A01-A06, while the evaluation set consists of unseen systems A07-A19. For CtrSVDD (Zang et al., 2024), we train on singing voice synthesis and singing voice conversion methods A01-A08 and test on unseen methods A09-A14.

**Evaluation Metrics:** We use Equal Error Rate (EER) as the primary evaluation metric. EER is threshold-independent and is well suited to score-based detection problems. It is defined as the error rate at the operating point where the false acceptance rate equals the false rejection rate. EER summarizes the trade-off between these two error types into

---

[1]https://www.kaggle.com/datasets/carlosgdcj/genius-song-lyrics-with-language-information

[2]https://suno.com

[3]https://udio.com

*Table 1.* **Equal Error Rate (EER↓) on the FakeMusicCaps dataset averaged over five training subsets.** Non-zero-shot models are trained separately on each of the five subsets and evaluated on all subsets under cross-subset testing. [†] denotes full fine-tuning.

| Method | Zero-Shot | MusicGen | MusicLDM | AudioLDM2 | Stable Audio Open | Mustango | Avg. |
|---|---|---|---|---|---|---|---|
| ***Waveform-Based Methods*** | | | | | | | |
| AASIST (Jung et al., 2022) | ✗ | 31.13 | 32.91 | 28.04 | 33.64 | 37.93 | 32.73 |
| MERT-AASIST (Yizhi et al., 2023) | ✗ | 19.67 | 26.95 | 19.89 | 21.27 | 28.58 | 23.27 |
| MERT-AASIST[†] (Yizhi et al., 2023) | ✗ | 11.31 | 20.98 | 3.49 | 12.18 | 30.26 | 15.64 |
| W2V2-AASIST (Tak et al., 2022) | ✗ | 19.56 | 26.80 | 19.71 | 26.44 | 36.51 | 25.80 |
| W2V2-AASIST[†] (Tak et al., 2022) | ✗ | 7.78 | 20.87 | 2.87 | 6.66 | 19.13 | 11.46 |
| WPT-W2V2-AASIST (Xie et al., 2026) | ✗ | 10.84 | 27.31 | 4.62 | 10.44 | 34.84 | 17.61 |
| ***Spectrogram-Based Methods*** | | | | | | | |
| Spec-ViT (Dosovitskiy et al., 2021) | ✗ | 21.02 | 32.91 | 12.11 | 21.42 | 25.78 | 22.65 |
| Spec-ConvNeXt (Liu et al., 2022) | ✗ | 15.78 | 30.40 | 11.42 | 15.24 | 32.40 | 21.05 |
| SpecTTTra-$\alpha$ (Rahman et al., 2025) | ✗ | 11.60 | 31.45 | 7.24 | 10.29 | 27.56 | 17.63 |
| SpecTTTra-$\beta$ (Rahman et al., 2025) | ✗ | 13.27 | 31.64 | 7.82 | 12.94 | 27.64 | 18.66 |
| SpecTTTra-$\gamma$ (Rahman et al., 2025) | ✗ | 13.42 | 30.91 | 9.13 | 13.24 | 28.33 | 19.00 |
| MusicDET (ours) | ✓ | 5.64 | 6.55 | 2.36 | 3.82 | 4.18 | 4.51 |
| Class-Conditional MusicDET (ours) | ✗ | 1.67 | 0.15 | 0.22 | 2.40 | 0.04 | 0.89 |

*Table 2.* **Equal Error Rate (EER↓) on the SONICS dataset averaged over two training subsets.** Non-zero-shot models are trained separately on Suno V3.5 and Udio 130 subsets and evaluated on all subsets under cross-subset testing. [†] denotes full fine-tuning.

| Method | Zero-Shot | Suno V2 | Suno V3 | Suno V3.5 | Udio 32 | Udio 130 | Avg. |
|---|---|---|---|---|---|---|---|
| ***Waveform-Based Methods*** | | | | | | | |
| AASIST (Jung et al., 2022) | ✗ | 25.37 | 18.30 | 22.80 | 29.40 | 17.23 | 22.62 |
| MERT-AASIST (Yizhi et al., 2023) | ✗ | 16.27 | 16.30 | 19.34 | 25.30 | 17.70 | 18.98 |
| MERT-AASIST[†] (Yizhi et al., 2023) | ✗ | 43.36 | 16.67 | 18.80 | 39.10 | 26.54 | 28.89 |
| W2V2-AASIST (Tak et al., 2022) | ✗ | 19.77 | 12.44 | 16.90 | 18.90 | 15.54 | 16.71 |
| W2V2-AASIST[†] (Tak et al., 2022) | ✗ | 16.20 | 0.37 | 0.47 | 24.97 | 21.70 | 12.74 |
| WPT-W2V2-AASIST (Xie et al., 2026) | ✗ | 14.63 | 7.84 | 14.60 | 19.47 | 13.26 | 13.96 |
| ***Spectrogram-Based Methods*** | | | | | | | |
| Spec-ViT (Dosovitskiy et al., 2021) | ✗ | 0.43 | 0.50 | 0.44 | 3.80 | 1.00 | 1.23 |
| Spec-ConvNeXt (Liu et al., 2022) | ✗ | 21.37 | 20.90 | 22.84 | 24.50 | 2.44 | 18.41 |
| SpecTTTra-$\alpha$ (Rahman et al., 2025) | ✗ | 0.70 | 1.34 | 0.93 | 7.83 | 2.50 | 2.66 |
| SpecTTTra-$\beta$ (Rahman et al., 2025) | ✗ | 1.90 | 3.00 | 3.10 | 8.27 | 3.84 | 4.02 |
| SpecTTTra-$\gamma$ (Rahman et al., 2025) | ✗ | 3.60 | 3.30 | 3.80 | 14.37 | 4.10 | 5.83 |
| MusicDET (ours) | ✓ | 2.80 | 3.20 | 2.93 | 2.73 | 2.80 | 2.89 |
| Class-Conditional MusicDET (ours) | ✗ | 0.00 | 0.00 | 0.00 | 0.00 | 0.00 | 0.00 |

a single value, and lower EER indicates stronger discriminative performance. It is widely used in tasks (Zhu et al., 2024; Ren et al., 2025) that require thresholded decisions, including forgery detection and speaker verification.

**Implementation Details:** For preprocessing, we resample all audio to 16 kHz and convert it to mono. We standardize each clip to 4 seconds by randomly cropping longer recordings and zero-padding shorter ones. We compute energy spectrogram features using the short-time Fourier transform (STFT) (Oppenheim et al., 1999) with `n_fft = 512`, `hop_length=160`, and `win_length=512`. SpecAugment (Park et al., 2019) is also applied during training by randomly masking time and frequency regions to improve generalization. We train all models for 10 epochs using Adam with an initial learning rate of $5 \times 10^{-4}$. For Mu-

sicDET, we set the batch size to 64, the number of flow steps $K$ in each band-wise flow to 2, and the number of frequency bands to 2. We use a Gaussian prior for real music with mean $\mu_{\text{real}} = 5$ and covariance $I$. In the class-conditional setting, we additionally use a class-conditional Gaussian prior for AI-generated music with mean $\mu_{\text{fake}} = -5$ and covariance $I$. All experiments run on a single NVIDIA RTX 4090 GPU with 24 GB of memory.

### 4.2. Experimental Results

**Analysis of Experimental Results:** We train each non–zero-shot model on a single TTM-specific subset of FakeMusicCaps and SONICS, and evaluate it across all subsets. Tables 1 and 2 report the mean EER, averaged over

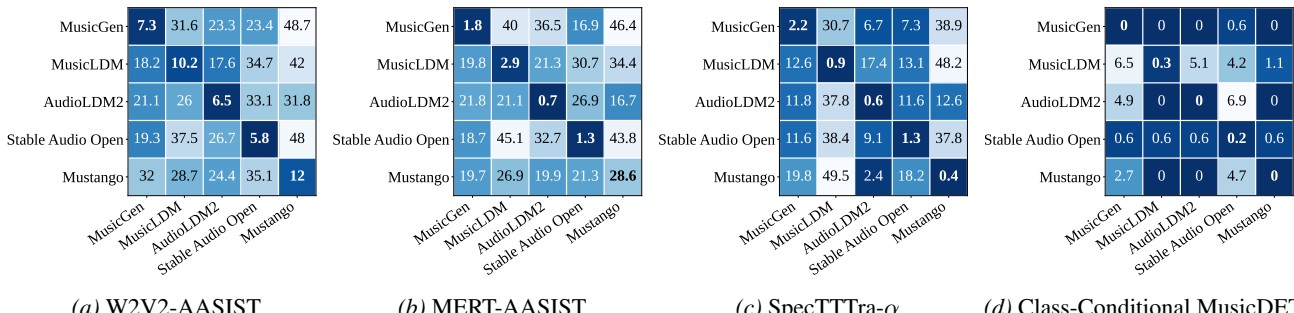

*(a) W2V2-AASIST*  *(b) MERT-AASIST*  *(c) SpecTTTra-α*  *(d) Class-Conditional MusicDET*

*Figure 4.* **Comparison of cross-generator generalization.** We select W2V2-AASIST, MERT-AASIST, and SpecTTTra-α as representative countermeasures. Each model is trained on one of the five specific subsets of FakeMusicCaps and evaluated on all subsets. In each confusion matrix, the vertical axis denotes the training subset and the horizontal axis denotes the test subset.

different training subsets, on FakeMusicCaps and SONICS, respectively. Note that MusicDET is the only zero-shot method in our comparison, as it is trained using real music only. To the best of our knowledge, there are currently no approaches specifically designed for cross-generator AI-generated music detection. We therefore adopt state-of-the-art audio and music forensics detection architectures as baselines, including AASIST, W2V2-AASIST, MERT-AASIST, WPT-W2V2-AASIST, and the SpecTTTra family.

As shown in Table 1, MERT-AASIST and W2V2-AASIST outperform AASIST by 9.46% and 6.93%, respectively. Wavelet Prompt Tuning (WPT) further reduces the EER of W2V2-AASIST by 8.19%. With full fine-tuning, performance can be improved further, albeit at the cost of higher computational overhead. Across generators, different SpecTTTra variants yield comparable performance, which generally lies between waveform-based methods with a frozen front end and those with full front-end fine-tuning. Overall, our proposed MusicDET maintains a clear advantage even in the zero-shot setting, outperforming all baseline detectors. Moreover, after incorporating AI-generated music via a class-conditional prior, MusicDET improves further and reduces the EER to 0.89%. As shown in Table 2, methods based on self-supervised feature extractors exhibit unstable performance and lag notably behind spectrogram-based approaches. MusicDET trained using only real music achieves performance comparable to SpecTTTra-β. Interestingly, the class-conditional variant of MusicDET correctly identifies all AI-generated music.

**Analysis of Cross-Generator Generalization:** The cross-generator generalization results (EER) are visualized in Figure 4, where the rows and columns correspond to the training and test subsets, respectively. The main diagonal corresponds to the close-set results, while the upper and lower triangular regions represent the open-set results. When the training and test subsets coincide, all methods achieve reasonably strong performance. However, competing methods generalize poorly to unseen generators. When

*Table 3.* **Efficiency Comparison between MusicDET and competing methods.** [†] denotes full fine-tuning.

| Method | Speed (M/S)↑ | FLOPs (G)↓ | Memory (GB)↓ | Param. (M)↓ | EER (%)↓ |
|---|---|---|---|---|---|
| *Waveform-Based Methods* | | | | | |
| AASIST | 271 | 9.62 | 0.20 | 0.30 | 32.73 |
| MERT-AASIST | 175 | 73.20 | 1.33 | 0.45 | 23.27 |
| MERT-AASIST[†] | 173 | 73.20 | 3.68 | 315.88 | 15.64 |
| W2V2-AASIST | 157 | 73.23 | 1.33 | 0.45 | 25.80 |
| W2V2-AASIST[†] | 158 | 73.23 | 3.68 | 315.89 | 11.46 |
| WPT-W2V2-AASIST (Xie et al., 2026) | 140 | 76.29 | 1.33 | 0.69 | 17.61 |
| *Spectrogram-Based Methods* | | | | | |
| Spec-ViT | 807 | 1.08 | 0.21 | 16.69 | 22.65 |
| Spec-ConvNeXt | 805 | 1.47 | 0.34 | 27.82 | 21.05 |
| SpecTTTra-α (Rahman et al., 2025) | 810 | 2.85 | 0.33 | 16.83 | 17.63 |
| SpecTTTra-β (Rahman et al., 2025) | 839 | 1.14 | 0.21 | 16.99 | 18.66 |
| SpecTTTra-γ (Rahman et al., 2025) | 846 | 0.74 | 0.21 | 17.17 | 19.00 |
| MusicDET | 516 | 4.09 | 0.11 | 8.13 | 4.51 |

trained on MusicGen, W2V2-AASIST, MERT-AASIST, and SpecTTTra-α often misclassify AI-generated music as real because they overfit to generator-specific artifacts and fail to generalize, resulting in high EERs of 48.7%, 46.4%, and 38.9%, respectively. In contrast, our method cleanly separates the two classes.

**Analysis of Efficiency:** We further evaluate practical deployability from the perspective of computational efficiency, reporting inference throughput (Speed, in music/s, denoted as M/S), floating-point operations (FLOPs), GPU memory consumption (Memory), and the number of trainable parameters (Param.) as shown in Table 3. Specifically, we load pre-processed music files during benchmarking to ensure that all methods are compared under identical input conditions. All experiments are conducted on a single NVIDIA RTX 4090 using forward passes with a batch size of 12. Fine-tuning self-supervised feature extractors can substantially reduce EER in cross-generator settings. However, these gains typically come with a larger trainable parameter budget, highlighting a non-trivial performance–efficiency trade-off. In contrast, MusicDET achieves the lowest EER with the smallest number of trainable parameters and competitive inference speed, delivering strong detection performance while maintaining high computational efficiency.

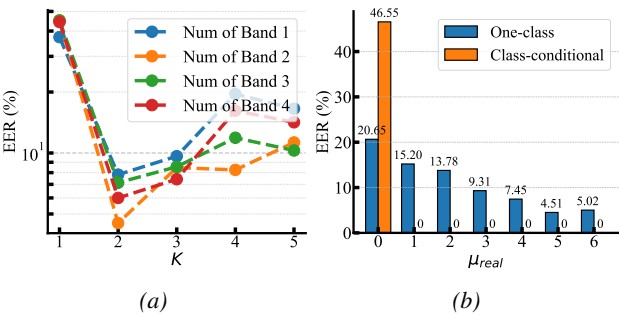

*Figure 5.* **Hyperparameter Analysis.** (a) Effect of the number of frequency bands and the depth of the band-wise normalizing flows. (b) Effect of the prior mean $\mu_{\text{real}}$ in the normalizing flows.

*Table 4.* **Leave-one-subdomain-out evaluation of MusicDET.** Jazz and piano are excluded from training, respectively.

| Generator | jazz (EER %) | piano (EER %) |
|---|---|---|
| MusicGen (Copet et al., 2023a) | 2.9 | 5.1 |
| MusicLDM (Chen et al., 2024a) | 4.4 | 7.3 |
| AudioLDM2 (Liu et al., 2024) | 1.5 | 2.2 |
| Stable Audio Open (Evans et al., 2025) | 1.5 | 2.9 |
| Mustango (Melechovsky et al., 2024) | 2.2 | 2.9 |
| Avg. | 2.5 | 4.1 |

### 4.3. Ablation Studies and Analysis

**Effect of Flow Steps $K$ and Number of Frequency Bands:**
We investigate the impact of the number of frequency bands and the number of normalizing flow steps $K$ per branch on MusicDET's performance. Figure 5a shows the log-scaled EER averaged over the five subsets of FakeMusicCaps, where the y-axis represents $\log(\text{EER})$. Across all values of $K$, employing multiple frequency bands consistently results in lower EER compared to a single-band configuration, highlighting the benefit of capturing frequency-specific statistical properties. Considering both accuracy and computational cost, we set the default configuration to two frequency bands and two flow steps per band.

**Effect of the Prior Mean $\mu$:** We further analyze the effect of the prior mean $\mu$ in MusicDET, as illustrated in Figure 5b. For one-class MusicDET, the reported EER is averaged over the five generator-specific subsets of FakeMusicCaps. In contrast, the class-conditional variant is trained on Music-Gen and evaluated on Mustango. For the one-class model, EER consistently decreases as $\mu_{\text{real}}$ increases. We attribute this trend to the fact that shifting the prior mean increases the latent-space separation between real and AI-generated samples, reducing their overlap in likelihood and thereby improving the discriminative power of the normalizing flow. An additional observation is that, in the class-conditional setting, the model can effectively distinguish real music from AI-generated music when using distinct priors for each class. In our experiments, we set $\mu_{\text{fake}} = -\mu_{\text{real}}$.

*Table 5.* **Performance comparison with state-of-the-art methods on the ASVspoof 2019 LA and CtrSVDD datasets.** All self-supervised front-end models are frozen.

| Method | ASVspoof 2019 LA | CtrSVDD |
|---|---|---|
| Spec-ResNet (He et al., 2016) | 5.58 | 22.59 |
| AASIST (Jung et al., 2022) | 1.48 | 11.54 |
| MERT-AASIST (Yizhi et al., 2023) | 4.80 | 12.55 |
| WavLM-AASIST (Chen et al., 2022) | 2.49 | 18.74 |
| W2V2-AASIST (Tak et al., 2022) | 1.28 | 8.23 |
| W2V2-ResNet (Gohari et al., 2025) | 2.07 | 11.72 |
| W2V2-LCNN (Gohari et al., 2025) | 3.63 | 12.78 |
| SingGraph (Chen et al., 2024b) | - | 6.23 |
| W2V2-MusicDET (ours) | 0.86 | 6.74 |

*Table 6.* **Robustness under different audio manipulations.** The EER differences relative to the untransformed setting are reported in parentheses for reference.

| Manipulation | MusicDET | Class-Conditional MusicDET |
|---|---|---|
| Pitch Shifting | 44.73 (↑40.22) | 44.35 (↑43.46) |
| Time Stretching | 2.44 (↓2.07) | 0.86 (↓0.03) |
| Equalization | 6.04 (↑1.53) | 1.27 (↑0.38) |
| Reverberation | 4.04 (↓0.47) | 0.90 (↑0.01) |
| White Noise | 44.11 (↑39.6) | 42.98 (↑42.09) |
| MP3 64kB/s | 41.75 (↑37.24) | 22.36 (↑21.47) |
| AAC 64kB/s | 35.85 (↑31.34) | 23.03 (↑22.14) |
| Opus 64kB/s | 22.15 (↑17.64) | 23.45 (↑22.56) |

**Analysis of Subdomain Generalization:** To examine the robustness of MusicDET under distribution shifts, we conduct a leave-one-subdomain-out evaluation, as shown in Table 4. During training, one subdomain is excluded at a time, including a specific genre (jazz) or an instrument-focused subset (piano). The model is then evaluated on corresponding subdomain-specific test sets containing both real and AI-generated music. The results, reported in terms of EER, demonstrate that MusicDET maintains strong performance even on unseen subdomains. These findings suggest that the model captures the underlying statistical structure of real music, rather than relying on subdomain-specific cues, highlighting its potential for robust zero-shot detection across diverse musical content.

**Transferability Across Audio Tasks:** As reported in Table 5, we evaluate the transferability of MusicDET to other audio domains on ASVspoof 2019 LA (Todisco et al., 2019) and CtrSVDD (Zang et al., 2024). Most state-of-the-art systems for ASVspoof follow a front-end/back-end design. Under the same paradigm, MusicDET achieves the best performance. It can be readily adapted to general ADD by using normalizing flows as the back end, operating on representations extracted by a wide range of front ends. On CtrSVDD, using the same acoustic features, MusicDET outperforms AASIST, ResNet, and LCNN by 1.49%, 4.98%, and 6.04% EER, respectively. It is surpassed only by Sing-Graph by 0.51%, a model specifically designed for this task.

**Robustness to Audio Manipulations:** Robustness to distribution shifts is essential for practical AI-generated music detection because creators rarely publish raw model outputs and often apply post-processing (Afchar et al., 2025). Following this motivation, we evaluate MusicDET under a set of common user-level audio manipulations, including random pitch shifting (±2 semitones), time stretching (80–120%), equalization, reverberation, additive white noise, and codec re-encoding at 64 kbps (MP3, AAC, and Opus). This protocol assesses robustness under realistic post-processing conditions (Afchar et al., 2025). We attribute this drop to the strong disturbance these manipulations impose on the spectral evidence the detector relies on, with pitch shifting altering harmonic structure, noise masking energy patterns, and re-encoding reducing bandwidth.

## 5. Conclusion

In this paper, we introduce a zero-shot setting for AI-generated music detection and present MusicDET, a normalizing-flow-based detector that learns the distribution of real music for generator-agnostic inference. Extensive experiments on FakeMusicCaps and SONICS show that MusicDET achieves state-of-the-art performance under cross-generator evaluation protocols. We further improve generalization by introducing a class-conditional prior when generated samples are available, and our analyses indicate that MusicDET is efficient for practical deployment and transfers beyond music to broader audio detection scenarios. We view improving robustness to real-world post-processing and adversarial attacks as an important direction for future research on AI-generated music detection.

## Acknowledgements

This work was supported in part by the National Science Foundation of China (62302093, 52441503), Jiangsu Province Natural Science Fund (BK20230833), the grant of the CIPS-SMP-Zhipu Large Model Fund, and the Big Data Computing Center of Southeast University.

## Impact Statement

This work aims to improve the detection of AI-generated music, which may help preserve content authenticity and support the responsible use of generative music technologies. At the same time, automatic detectors may produce false positives or false negatives, and their outputs should not be used as the sole basis for high-stakes decisions affecting artists, users, or content distribution. We therefore encourage the use of AI-generated music detectors as decision-support tools, together with human review, transparent reporting, and continuous evaluation on diverse and evolving datasets.

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

# A. Appendix

## A.1. Preliminaries of Normalizing Flows

Normalizing flows are likelihood-based generative models that enable exact density estimation via a sequence of invertible transformations between the data space and a latent space with a tractable prior distribution. By explicitly modeling the distribution of in-domain data, NFs naturally assign low likelihood values to out-of-distribution (OOD) or anomalous samples, making them particularly suitable for detection tasks.

In the zero-shot AI-generated music detection setting, we define the spectral energy representation of real music as the data space $\mathcal{X}$ and the corresponding latent space as $\mathcal{Z}$. Let $x \in \mathcal{X}$ denote a spectral feature extracted from a music signal, and let $z \in \mathcal{Z}$ be its latent representation. A normalizing flow defines a bijective mapping

$$z = f_\theta(x), \qquad x = f_\theta^{-1}(z), \tag{12}$$

where $f_\theta : \mathcal{X} \leftrightarrow \mathcal{Z}$ is parameterized by $\theta$. The latent variable $z$ follows a base distribution $p_Z(z)$, typically a Gaussian. In practice, $f_\theta$ is implemented as a composition of $K$ invertible transformations:

$$f_\theta = f_K \circ f_{K-1} \circ \cdots \circ f_1, \tag{13}$$

with intermediate states $h_0 = x$ and $h_j = f_j(h_{j-1})$ for $j = 1, \ldots, K$. Given a base density $p_Z(z)$, the data likelihood can be computed exactly using the change-of-variables formula (Villani, 2003):

$$p_X(x) = p_Z\big(f_\theta(x)\big) \left| \det J_{f_\theta}(x) \right|, \tag{14}$$

where $J_{f_\theta}(x)$ denotes the Jacobian matrix of $f_\theta$ with respect to $x$. Taking logarithms yields

$$\log p_X(x) = \log p_Z(h_K) + \sum_{j=1}^{K} \log \left| \det J_{f_j}(h_{j-1}) \right|. \tag{15}$$

To ensure computational efficiency, each transformation $f_j$ is designed such that its Jacobian determinant can be evaluated tractably, as in coupling-based flow architectures. Assuming a standard Gaussian prior $p_Z(z) = \mathcal{N}(0, I)$ in a $d$-dimensional latent space, the log-likelihood admits the explicit form

$$\log p_X(x) = -\frac{d}{2} \log(2\pi) - \frac{1}{2} \| h_K \|_2^2$$
$$+ \sum_{j=1}^{K} \log \left| \det J_{f_j}(h_{j-1}) \right|. \tag{16}$$

This exact likelihood serves as the fundamental decision score for AI-generated Music detection.

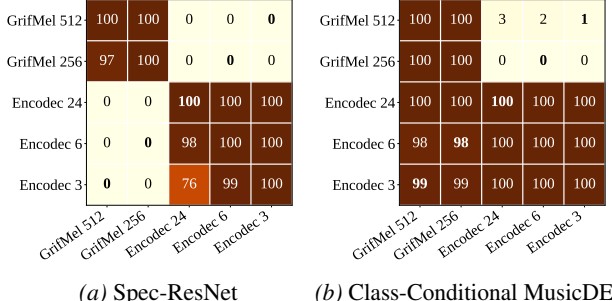

*(a)* Spec-ResNet      *(b)* Class-Conditional MusicDET

*Figure 6.* **Confusion matrices for reconstructed music with overall accuracy.** (a) Spec-ResNet fails to transfer across reconstruction families. (b) Class-Conditional MusicDET achieves better generalization, where training on higher-quality reconstructions enables reliable detection on lower-quality reconstructions.

## A.2. Evaluation on EnCodec-Reconstructed Music

**Motivation.** Directly comparing *real* and *synthetic* music can be confounded by genre, instrumentation, and encoding differences. To better isolate generator artifacts, Afchar et al. (Afchar et al., 2025) use a controlled reconstruction-based evaluation. Starting from the same real-music pool, they create reconstructed versions with different decoders. Originals and reconstructions share identical musical content and can be stored under matched codec and bitrate settings, which encourages detectors to focus on decoder-induced artifacts rather than dataset bias.

**Dataset and Protocol.** We construct a benchmark from the FMA-medium dataset (Defferrard et al., 2017) using 25,000 real music clips. For each clip, we generate reconstructed versions using (i) EnCodec (Défossez et al., 2023) at bitrates of 3, 6, and 24 kbps, and (ii) GrifMel (AlBadawy et al., 2022), a Griffin–Lim-based mel inversion pipeline with 256 and 512 mel bins. Each reconstructed sample is paired with its corresponding original, forming a *real vs. reconstructed* discrimination task. We then train and evaluate class-conditional MusicDET (a) under within-family transfer across EnCodec bitrates and GrifMel mel bins; and (b) under cross-family transfer between EnCodec and GrifMel.

**Results.** Figure 6 shows the confusion matrices of Spec-ResNet (Afchar et al., 2025) and MusicDET under cross-generator evaluation, where the y-axis denotes the training subset and the x-axis denotes the test subset. In the *within-family* setting, both methods transfer well across EnCodec operating points and GrifMel mel bins. In the *cross-family* setting, Spec-ResNet generalizes poorly beyond its training family. When trained on EnCodec, our method reliably detects GrifMel samples, but not vice versa. We attribute this asymmetry to reconstruction fidelity: EnCodec outputs are perceptually closer to the originals and thus contain subtler artifacts that are harder to learn and less transferable.

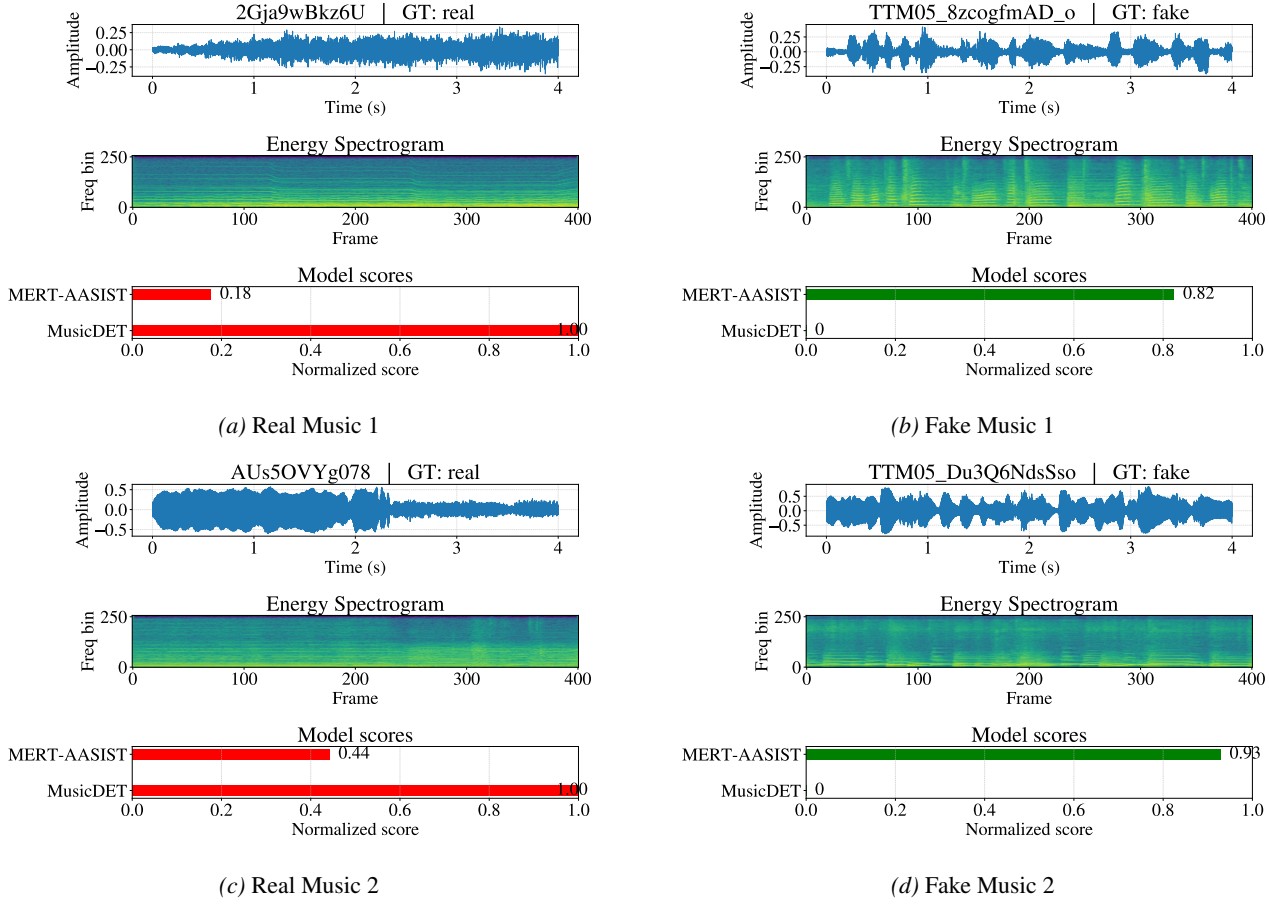

*(a)* Real Music 1

*(b)* Fake Music 1

*(c)* Real Music 2

*(d)* Fake Music 2

*Figure 7.* **Visualization of MERT-AASIST and MusicDET predictions on real and AI-generated music samples.** The left column corresponds to real samples, while the right column shows fake counterparts. For each sample, we visualize the raw waveform, the energy spectrum, and the corresponding model prediction results.

### A.3. Visualizations of Music Samples

As illustrated in Figure 7, we present the prediction outcomes for two real music samples (left column) and two AI-generated music samples (right column). All models are trained on MusicGen (Copet et al., 2023a) and evaluated on Mustango (Melechovsky et al., 2024), following a cross-model generalization setting. MusicDET is a likelihood-based density modeling method that produces log-likelihood scores rather than explicit probabilities. Consequently, binary decisions are obtained by applying a manually selected log-likelihood threshold of $-20$. In contrast, MERT-AASIST outputs **posterior probabilities indicating whether a given sample is real.** As shown in the figure, MERT-AASIST fails to correctly classify all four samples, assigning low confidence scores to real music while yielding overly confident predictions for AI-generated samples. By comparison, MusicDET correctly discriminates between real and AI-generated music in all cases, indicating a stronger robustness to subtle generation artifacts and improved generalization across generative models.

