# OpenReview forum: "MusicDET: Zero-Shot AI-Generated Music Detection"
_ICML.cc/2026/Conference — ICML 2026 regular_

### Official Review · Reviewer_NidG · 2026-03-04

**Soundness:** 3
**Presentation:** 4
**Significance:** 3
**Originality:** 3
**Overall Recommendation:** 5
**Confidence:** 3

**Summary:**

This paper addresses zero-shot AI-generated music detection. Instead of depending on artifacts tied to particular generators, the method models the distribution of real music and uses it for detection. The authors develop this idea into a likelihood-based framework, MusicDET, and also consider a class-conditional variant for the setting where fake samples are available. The experiments indicate good performance, especially in terms of generalization to unseen generators. Overall, the paper studies a relevant problem and the proposed approach appears reasonably effective.

**Compliance With Llm Reviewing Policy:**

Affirmed.

**Final Justification:**

The rebuttal satisfactorily addresses my major concerns through clearer motivation, stronger empirical validation, and the inclusion of previously missing implementation details.

**Key Questions For Authors:**

(1) The proposed method mainly relies on likelihood modeling based on normalizing flows to distinguish between real music and AI-generated music. Could the authors further explain or analyze why this approach can reliably separate these two types of samples in this task? It would also be helpful if the authors could provide additional theoretical or statistical insights to support this observation.

(2) In the experiments, instrumental music and music containing vocal singing are treated uniformly without explicit distinction. Have the authors analyzed the potential impact of vocal music on the detection performance? Would the model exhibit different performance when evaluated separately on purely instrumental music or music containing vocals?

(3) On the SONICS dataset, the reported EER of the Class-Conditional MusicDET is close to zero, which is an extremely strong result. Could the authors clarify whether the experiment was repeated multiple times or whether any statistical stability analysis (e.g., results under different training splits) was conducted to verify the robustness of this result?

(4) The paper provides most of the experimental parameters, but some training details (such as the number of training epochs) are not fully specified. Could the authors provide these additional implementation details, or do they plan to release the code after publication to further improve the reproducibility of the work?

(5) Could the authors also report the detection performance of the proposed method on more recent music generation models, such as Suno v5 or the ACE-Step series? Evaluating on these newer models would help further demonstrate the generalization ability of the detector in more up-to-date generation scenarios.

**Limitations:**

No. The paper mainly focuses on method design and experimental validation, with limited discussion of the method’s limitations and potential societal impacts. For example, the paper does not sufficiently analyze the applicability of the model to different types of music (such as purely instrumental versus vocal music), nor does it discuss potential limitations in more realistic and complex environments (e.g., audio compression or post-processing). It would be beneficial for the authors to include a more systematic discussion of the method’s limitations and potential impacts in order to make the paper more complete.

**Strengths And Weaknesses:**

Soundness: The paper is sound overall and the experimental results are strong. The comparisons are reasonably comprehensive, and the evaluation setup seems appropriate. My main concern is that the paper relies heavily on empirical evidence. The method is based on likelihood modeling with normalizing flows, but the paper does not really explain why this should reliably distinguish real music from AI-generated music. I also think the treatment of instrumental and vocal music deserves more discussion. These two cases may have different acoustic characteristics, and it is not clear whether that affects performance. In addition, the near-zero EER reported for class-conditional MusicDET on SONICS is unusually strong, so it would help to know whether this was consistent across repeated runs or supported by any statistical analysis. The experimental evaluation does not include some of the most recent text-to-music generation models such as Suno v5 or the ACE-Step series, which have been released in the past two years. Including these models could better demonstrate the generalization of MusicDET under truly current cross-generator scenarios. As such, the reported results may not fully reflect performance against the latest generation methods.

Presentation: The paper is readable and generally well organized. The motivation and main idea are clear, and the figures and tables are useful. Still, some implementation details are missing, especially in the training setup. Adding those details would improve reproducibility, and code release would help as well.

Significance: AI-generated music detection is an important problem, and the cross-generator setting makes the work more relevant in practice. The paper has value from that perspective.

Originality: The paper brings a somewhat different angle to this problem by using probability distribution modeling and a likelihood-based detection framework. The frequency-wise design also adds some novelty. I would view the originality as moderate but reasonable.

---

> ### Author Rebuttal · Authors · 2026-03-30
>
> We thank the reviewer for the time and the positive comments on the organization of our work. Below we respond to the questions.
>
> > Q1. Could the authors further explain or analyze why this approach can reliably separate these two types of samples in this task?
>
> Our method is based on modeling the statistical distribution of real music in the time–frequency energy domain. The key intuition is that real music arises from **physical acoustic processes and human compositional mechanisms, which impose strong structural constraints** on the signal, such as stable harmonic structures, cross-band energy consistency, and coherent temporal evolution. As a result, from a statistical perspective, real music tends to lie in a well-structured and relatively concentrated region of the feature space. Although AI-generated music is often perceptually realistic, current generative models are not explicitly designed to reproduce the full physical and statistical regularities of real music. Consequently, generated samples may still exhibit systematic discrepancies in harmonic organization, inter-band energy coupling, and temporal continuity (as illustrated in Figure 1 of the manuscript).
>
> From a modeling perspective, our approach follows a likelihood-based anomaly detection framework. The normalizing flow learns the distribution of real music, so samples that deviate from these regularities receive lower likelihoods and can thus be distinguished from authentic ones. **The energy spectrogram representation further facilitates this process by emphasizing perceptually relevant structural patterns while reducing irrelevant variability**, making the distribution of real music easier to model. Empirically, we observe a clear likelihood separation between real and generated music across multiple datasets and unseen generators, which provides empirical support for our hypothesis that generated music behaves as out-of-distribution data under the learned real-music distribution.
>
> > Q2. Have the authors analyzed the potential impact of vocal music on the detection performance?
>
> We analyze the impact of vocal content on detection performance. Specifically, we construct a balanced evaluation set on YuE [1] and ACE-Step [2], including male vocals, female vocals, and instrumental music (184/183/183 samples). MusicDET achieves low EERs of 6.6% and 0.6% on the two generators, respectively, when evaluated on the combined set.
> Although these results are reported on the aggregated data, the consistently low EER on a balanced dataset suggests **no evident performance bias between vocal and instrumental music.**
>
> This behavior can be explained by our modeling objective. Although vocal and instrumental music differ in timbre and structure, they share fundamental spectral statistical properties, such as harmonic patterns, energy distributions, and cross-band correlations. MusicDET focuses on these underlying regularities rather than semantic content, enabling robust performance across different audio types.
>
> > Q3. Could the authors clarify whether the experiment was repeated multiple times or whether any statistical stability analysis was conducted to verify the robustness of this result?
>
> Class-conditional MusicDET is trained separately on Suno v3.5 and Udio 130, and the final result is averaged over the two runs. We further repeat the experiments with different random seeds and obtain highly consistent results, indicating that **this performance is stable and not due to randomness.** We attribute this strong performance to two factors: the high quality and diversity of real music in SONICS, and the enhanced likelihood-space separation brought by class-conditional modeling.
>
> > Q4. Could the authors provide these additional implementation details?
>
> During training, both cropping and padding are performed randomly, whereas during evaluation, center cropping and center padding are used to ensure deterministic preprocessing. We also apply per-sample RMS normalization to the waveform before STFT. The model is trained using Adam (β₁ = 0.9, β₂ = 0.999, ε = 1 × 10⁻⁸) with a weight decay of 5 × 10⁻⁴. In addition, gradient clipping with a maximum norm of 100 is applied to stabilize training. **To further improve reproducibility, we will release the full codebase.**
>
> > Q5. Could the authors also report the detection performance of the proposed method on more recent music generation models, such as Suno v5 or the ACE-Step series?
>
> Since Suno v5 is not open-sourced, we instead evaluate MusicDET on two recent music generation models, YuE [1] and ACE-Step [2], to further assess its generalization ability. We conduct additional experiments accordingly, and we refer the reviewer to our response to Reviewer qtWt (Q1) for detailed results and analysis.
>
> [1] YuE: Scaling Open Foundation Models for Long-Form Music Generation
>
> [2] ACE-Step: A Step Towards Music Generation Foundation Model

---

> > ### Author Rebuttal · Reviewer_NidG · 2026-04-01
> >
> > The rebuttal addresses my main questions in a satisfactory way. The authors clarified the intuition behind the likelihood-based approach, provided additional analysis on instrumental and vocal music, explained the stability of the SONICS result across repeated runs, and added the missing implementation details. I also appreciate the added evaluation on more recent generators, which strengthens the empirical support for the paper. Overall, I consider my concerns adequately addressed, and my overall assessment remains unchanged.

---

> > > ### Author Response · Authors · 2026-04-02
> > >
> > > Thank you very much for your time and effort in reviewing our paper. We sincerely appreciate your positive comments and encouraging feedback.

---

### Official Review · Reviewer_RES4 · 2026-03-09

**Soundness:** 4
**Presentation:** 4
**Significance:** 3
**Originality:** 3
**Overall Recommendation:** 4
**Confidence:** 5

**Summary:**

The paper introduce a novel method for detecting in zero shot manner AI generated music. The proposed detector train only on real music without any generated music. The paper proposed MusicDET, a detection method based on normalizing flows that model the probability of real music features. The proposed method effectively detect out of distribution music samples and the proposed method evaluated on the FakeMusicCaps and SONICS datasets

**Compliance With Llm Reviewing Policy:**

Affirmed.

**Final Justification:**

The rebuttal addressed my main concerns, my evaluation keep unchanged - 4: Weak accept

**Key Questions For Authors:**

1. The proposed method takes the output of the normalization flow and with a fixed threshold makes a binary decision. There is no explanation of how this threshold is determined. Please provide explanation how you set this threshold?
2. The results get worse when there is pitch shift or white noise, why is this happening?

**Limitations:**

--

**Strengths And Weaknesses:**

Strength:
1. The paper introduces a novel architecture which uses normalization flaw to detect ai generated music, furthermore, the proposed method has frequency wise decomposition to detect different features such as timber and rhythm
2. The method trains only on real music samples which leads to an agnostic generator model. This idea fixes a major flaw in current methods that detect ai generated music samples.
3. The proposed method, MusicDET, achieves the best results compared to the previous method with only 8.1M parameters while previous methods like W2V2-AASIST has 315M parameters. moreover the proposed method can be deployed in real time scenario

Weakness:
1. The proposed method takes the output of the normalization flow and with a fixed threshold makes a binary decision. There is no escalation of how this threshold is determined.
2. The results get worse when there is pitch shift or white noise, why is this happening?
3. The proposed method use low level features such as energy spectrogram and don't pay attention to long term structural

---

> ### Author Rebuttal · Authors · 2026-03-30
>
> We would like to thank the reviewer for taking the time to review our work and for the positive comments regarding its novelty and significance. Below are our responses to the comments in Weaknesses and Questions.
>
> > W1Q1. There is no explanation of how this threshold is determined. Please provide explanation how you set this threshold?
>
> In our current implementation, the decision threshold is not a carefully tuned constant tied to a specific test set. Instead, it can be selected in a data-driven manner based on the likelihood distribution of real training samples. A natural choice is a quantile-based thresholding scheme, where the threshold is determined by a target acceptance rate for real music. This strategy is simple, reproducible, and well-aligned with the one-class nature of our method, as the model is designed solely to characterize the distribution of real music. We clarify this selection in the revised manuscript, making the paper more complete and self-contained. We further evaluate the detector across thresholds around the operating point of -20 used in the paper. As shown below, accuracy remains above 98% over a wide range, indicating limited sensitivity to threshold variation and no reliance on a finely tuned operating point.
> |Threshold|-10|-15|-20|-25|-30|
> |-|-|-|-|-|-|
> |Acc. (%)|98.1|100|99.9|99.8|99.7|
>
> > W2Q2. The results get worse when there is pitch shift or white noise, why is this happening?
>
> We sincerely thank the reviewer for raising insightful questions regarding robustness. As shown in Table 5 of the manuscript, our method exhibits a noticeable performance drop under perturbations such as pitch shifting, white noise, and low-bitrate compression (MP3/AAC/Opus), while remaining relatively stable under time stretching, equalization, and reverberation. This observation is consistent with prior studies [1][2][3], which report that **most machine learning-based forgery detection methods lack robustness to such perturbations when out-of-distribution shifts are not explicitly addressed**. From a modeling perspective, our approach leverages normalizing flows to learn the spectral distribution of real, clean music. Perturbations such as pitch shifting alter harmonic positions and frequency alignment globally, white noise introduces random energy across the entire frequency spectrum, and low-bitrate compression degrades high-frequency details while often introducing compression artifacts. These effects cause real music itself to deviate from the learned distribution, thereby weakening the discriminative power of likelihood-based detection.
>
> We also explore incorporating perturbed real music into training to expand the support of the real-data distribution. However, our experiments indicate that such straightforward data augmentation does not effectively mitigate the performance degradation caused by these strong distribution shifts. One possible explanation is that these perturbations substantially affect both real music and AI-generated music, reducing their separability in the feature space and making it more difficult for the model to maintain stable discrimination.
>
> Therefore, we believe that this issue cannot be fully resolved by simple data augmentation alone. **A more promising direction may involve introducing dedicated robustness mechanisms**, such as explicitly learning perturbation-invariant representations or modeling distributional variations under different perturbation conditions. We view this as an important challenge for universal detection frameworks under distribution shifts. However, this is beyond the scope of the current work. We will clarify this point in the limitations section and consider it an important direction for future research.
>
> [1] AI-Generated Music Detection and its Challenges. ICASSP'2025
>
> [2] CNN-generated images are surprisingly easy to spot... for now. ICCV'2020
>
> [3] A large-scale challenging dataset for deepfake forensics. ICCV'2020
>
> > W3. The proposed method use low level features such as energy spectrogram and don't pay attention to long term structural
>
> We thank the reviewer for pointing out this limitation. We agree that the current method mainly relies on low-level energy spectrograms and does not explicitly model long-term musical structures, such as rhythm or global temporal dependencies. This design is motivated by our goal of authenticity detection rather than music understanding. Prior work and our experiments suggest that discrepancies between real and generated music are often reflected in local time-frequency statistics, which can be effectively captured by energy spectrograms. Nevertheless, we agree that incorporating multi-scale modeling to jointly capture local spectral statistics and long-term structure is a promising direction, and can be viewed as a potential extension of likelihood-based detection frameworks in future work.

---

> > ### Author Rebuttal · Reviewer_RES4 · 2026-04-02
> >
> > Thank you for addressing my concerns

---

> > > ### Author Response · Authors · 2026-04-03
> > >
> > > We sincerely thank the reviewer for the time and effort devoted to evaluating our paper. We are also encouraged to know that our previous response has helped clarify the reviewer’s concern. Regarding **robustness under different audio manipulations (W2Q2)**, we would like to provide one additional clarification.
> > >
> > > In our previous response, we explained that MusicDET faces particular challenges in achieving robustness to specific audio manipulations under the strict **zero-shot setting**, where only real samples are available for training. More recently, we found that **when fake samples are available during training, simple data augmentation can lead to noticeable improvements in class-conditional MusicDET**. The EER results are summarized below, where ↓ denotes the reduction in EER compared with the version trained without data augmentation:
> > >
> > > | Manipulation | Class-conditional MusicDET | SpecTTTra-α |
> > > |---|---:|---:|
> > > | Pitch Shifting | 33.67 (↓10.68) | 32.40 |
> > > | White Noise | 21.85 (↓21.13) | 21.82 |
> > > | AAC 64 kB/s | 7.82 (↓12.41) | 15.02 |
> > > | Opus 64 kB/s | 10.62 (↓12.83) | 20.07 |
> > >
> > > These results show that class-conditional MusicDET achieves performance comparable to that of the SOTA method SpecTTTra-α under pitch shifting and white noise perturbations, while achieving lower EERs under AAC and Opus compression attacks.
> > >
> > > Overall, these findings suggest that **robustness is particularly challenging in the strict zero-shot setting, while it can be substantially improved when the training condition is relaxed**. We hope this additional clarification is helpful, and we sincerely appreciate the reviewer’s insightful comments on this issue.

---

### Official Review · Reviewer_56G8 · 2026-03-10

**Soundness:** 3
**Presentation:** 3
**Significance:** 4
**Originality:** 4
**Overall Recommendation:** 6
**Confidence:** 4

**Summary:**

This paper presents MusicDET, a novel music generation technique that operates without seeing any generated audio during training. The authors propose using frequency-guided normalizing flows to map the energy spectrograms of real music into a tractable prior distribution. By calculating the likelihood of a test sample under this learned real-music distribution, the model can reliably discern real audio from generated audio. Empirically, MusicDET outperforms existing baselines in cross-generator scenarios and demonstrates high computational efficiency.

**Compliance With Llm Reviewing Policy:**

Affirmed.

**Final Justification:**

The proposed MusicDET framework introduces an original and sound approach. It not only establishes state-of-the-art performance in cross-generator music detection but also successfully demonstrates the architecture's impressive transferability to speech deepfake tasks. The methodology is presented with high clarity, proves to be computationally efficient, and represents a significant, highly practical step forward for real-world audio forensics where generative models continuously and rapidly evolve.

The authors' rebuttal was comprehensive and fully resolved my initial concerns, reinforcing my confidence in the work and prompting me to raise my overall recommendation to a Strong Accept. I particularly appreciate the newly conducted leave-one-subdomain-out experiments, which convincingly demonstrate that the framework captures the fundamental statistical regularities of real music rather than overfitting to specific styles, maintaining low error rates even when specific genres (jazz) or instruments (piano) are deliberately withheld during training. Furthermore, the authors provided thoughtful, nuanced discussions regarding the framework's behavior under severe audio manipulations and expertly addressed the theoretical constraints of adapting this spectral technique to the image domain. Given the strong empirical evidence, the rigorous soundness of the methodology, and the authors' diligent revisions, this paper stands out as an excellent contribution with high potential impact.

**Key Questions For Authors:**

1. Have you tried training MusicDET while explicitly withholding a specific genre or instrument, and then testing real samples from that excluded distribution? Would it be feasible to try that, and what false positive rates does it yield?
2. Given the severe performance drop under Pitch Shifting and Codec Compression (Table 5), how might the framework be adapted to be more robust? Could data augmentation (e.g., training the real-music distribution on pitch-shifted/compressed real audio) solve this, or would it make the real distribution too broad to accurately detect fakes?
3. Could this technique be adapted to the image domain? By computing the 2D FFT of an image, you could bring it into the spectral domain and then maybe reuse your normalizing flow technique in the same manner. Is this theoretically feasible, or are there audio-specific constraints preventing it?
4. The paper contains a couple of typos (e.g., in the paragraph at Section 3.2, line 170, there’s “usic” instead of “music”). Please fix!

**Limitations:**

Yes

**Strengths And Weaknesses:**

**Strengths:**
1. By training exclusively on real music, MusicDET avoids overfitting to the specific artifacts of any single generative model. This makes the framework highly sound and practical for real-world deployment against evolving generators.
2. The model consistently outperforms all other available baselines in open-set/cross-generator detection tasks, providing concrete proof of the method's efficacy.
3. Beyond just music, the authors successfully demonstrate the transferability of their method to speech deepfake detection (ASVspoof and CtrSVDD in Table 4), proving the general applicability of the architecture.

**Weaknesses:**
1. Since the entire technique is based on one-class density estimation of the real music distribution, it stands to reason that if a certain instrument, genre, or recording style is not present in the training set, it will be detected as unlikely by the model (yielding a false positive). The paper lacks an analysis of how the model handles rare or out-of-distribution real music.
2. While the robustness testing in Table 5 is appreciated, the results reveal a critical vulnerability to specific audio manipulations. To be fair, some manipulations like White Noise degrade the perceptual quality of the music anyway, making them a less realistic threat vector for an attacker. However, the model completely breaks against ±2 semitones Pitch Shifting and 64kbps MP3/AAC compression. These are massive real-world limitations, as pitch-shifted audio (e.g., Nightcore) and heavy compression are reasonable effects that may not necessarily sound "bad" to human listeners.

---

> ### Author Rebuttal · Authors · 2026-03-30
>
> We would like to thank the reviewer for taking the time to review our work and the positive comments regarding its novelty. Below are our responses to the comments in Weaknesses and Questions.
>
> > W1Q1. Have you tried training MusicDET while explicitly withholding a specific genre or instrument, and then testing real samples from that excluded distribution?
>
> We agree that this setup provides a more rigorous test of whether MusicDET has truly learned the general statistical properties of real music, rather than relying on specific styles or instrument cues present in the training data.
> We further conduct a leave-one-subdomain-out evaluation. Specifically, during training, we exclude one music subdomain at a time, including a specific genre (jazz) and an instrument-dominated subset (piano). During testing, we construct corresponding subdomain-specific test sets, each containing both real and AI-generated music from that subdomain, to evaluate the model under distribution shift. The results are summarized below (EER ↓):
> |Excluded|MusicGen|MusicLDM|AudioLDM2|Stable Audio Open|Mustango|Avg.|
> |-|-|-|-|-|-|-|
> |jazz|2.9|4.4|1.5|1.5|2.2|2.5|
> |piano|5.1|7.3|2.2|2.9|2.9|4.1|
>
> After excluding a specific genre or instrument, the EER remains low, with no clear degradation compared to the full-training setting (EER = 4.5%), indicating strong generalization under distribution shifts within real music.
>
> Since EER is measured at the operating point where the false positive rate equals the false negative rate, its stability suggests that the overall separability between real and generated music remains similar on the unseen subdomain, with no clear evidence of a substantial increase in false positives. We attribute this to the core design of MusicDET: rather than relying on style-specific or instrument-specific cues, it captures shared statistical regularities of real music in the time-frequency domain, enabling effective generalization to unseen subdomains.
>
> > W2Q1. Given the severe performance drop under Pitch Shifting and Codec Compression (Table 5), how might the framework be adapted to be more robust?
>
> We sincerely thank the reviewer for the insightful questions regarding robustness. Due to the space limit, we kindly refer the reviewer to our response to Reviewer RES4 (W2Q2), where we provide a more detailed analysis.
>
> > Q3. Could this technique be adapted to the image domain?
>
> We thank the reviewer for this insightful question. In principle, this idea could be extended to the image domain. More broadly, our method can be viewed as a frequency-domain perspective for distribution modeling and forgery detection. However, we would also like to clarify that, although the general intuition may transfer, audio spectrograms and image spectra are **fundamentally different due to domain-specific structures and assumptions.** An audio spectrogram is a structured time-frequency representation with strong temporal continuity, cross-band correlation, and harmonic patterns, which are explicitly exploited by our frequency-wise decomposition and flow-based modeling. In contrast, the 2D FFT spectrum of images has weaker semantic interpretability, and its statistics are heavily influenced by content, scene, and texture, making unified likelihood modeling less effective for capturing forgery-specific cues.
>
> Overall, we believe that relying on the frequency domain alone may not be sufficient for image forgery detection, since many generation artifacts are spatially localized. Even in industrial anomaly detection[1], normalizing flows are typically applied in the spatial feature space rather than the frequency domain, highlighting the importance of spatial representations for capturing localized artifacts. Accordingly, a more reasonable approach may be to treat the frequency-domain branch as a complementary component rather than the primary modeling pathway. In addition, image generation artifacts are often not globally uniform. Such artifacts are more likely to appear in local regions, near edges, within repetitive textures, or at specific scales. A possible extension, therefore, would be to extract local spectral features at the patch level and then use normalizing flows to model the distribution of local frequency/space statistics in real images. We believe this may help improve the modeling of fine-grained artifacts.
>
> [1] SANFlow: Semantic-Aware Normalizing Flow for Anomaly Detection and Localization. NeurIPS'2023.
>
> > Q4. The paper contains a couple of typos. Please fix!
>
> Thank you for carefully pointing this out. We have corrected the typo and thoroughly proofread the manuscript to ensure overall correctness.

---

> > ### Author Rebuttal · Reviewer_56G8 · 2026-04-02
> >
> > The authors fully addressed my comments, and the further experiments were much appreciated.

---

> > > ### Author Response · Authors · 2026-04-02
> > >
> > > We would like to express our sincere appreciation to the reviewers for their time and effort in reviewing our manuscript, as well as for their encouraging and insightful comments.

---

### Official Review · Reviewer_qtWt · 2026-03-13

**Soundness:** 3
**Presentation:** 3
**Significance:** 3
**Originality:** 3
**Overall Recommendation:** 4
**Confidence:** 4

**Summary:**

This paper addresses the challenge of detecting AI-generated music, specifically focusing on the performance degradation of discriminative detectors when encountering unseen generative models. The authors propose MusicDET, a zero-shot, generator-agnostic detection framework that models the distribution of real music features using frequency-guided normalizing flows. By learning an invertible mapping from real music spectral features to a simple prior distribution, the framework identifies AI-generated music as samples that deviate from the learned distribution, effectively enabling open-set detection without requiring any generated training samples. Experiments on FakeMusicCaps and SONICS datasets demonstrate that MusicDET consistently outperforms conventional discriminative detectors under cross-generator evaluation protocols.

**Compliance With Llm Reviewing Policy:**

Affirmed.

**Key Questions For Authors:**

The current evaluation covers several popular generative models; however, recent advanced music generation models such as YuE[1] or ACE-Step[2] are absent from your experiments. To better validate the "zero-shot" and "generator-agnostic" claims of MusicDET, could you provide detection results (EER) for these newer models?


[1] YuE: Scaling Open Foundation Models for Long-Form Music Generation
[2] ACE-Step A Step Towards Music Generation Foundation Model

**Limitations:**

No limitations discussed.

**Strengths And Weaknesses:**

Strengths

1. The paper introduces a well-motivated zero-shot setting for AI-generated music detection. By training exclusively on real music distributions using normalizing flows, the framework effectively bypasses the dependency on generator-specific training data, which is a significant step toward addressing the "catastrophic forgetting" or performance degradation often seen in discriminative detectors when encountering unseen models.

2. MusicDET provides a creative application of frequency-guided normalizing flows to the audio domain. The frequency-wise decomposition strategy, which allows the model to capture music-specific regularities across different spectral bands, is a technically sound method for modeling the complex and multimodal distribution of authentic music.

3. The empirical results consistently outperform strong baseline detectors across diverse datasets (FakeMusicCaps and SONICS). The model's ability to maintain high discriminative performance in cross-generator and open-set scenarios demonstrates its practical utility for real-world forensic applications.

Weaknesses

1. While the likelihood-based approach is theoretically clean, the paper relies on a "manually selected log-likelihood threshold" (e.g., −20) to make binary decisions. This thresholding strategy feels arbitrary and potentially sensitive to different musical genres or production styles. A more systematic, data-driven approach for threshold selection—or a discussion on its stability—would strengthen the practical deployment of the detector.

2. he method relies on a fixed transformation to STFT-based energy spectrograms. While this preserves temporal and spectral structure, it may overlook phase-based cues or deep temporal dependencies that other state-of-the-art deep learning-based detectors might capture. The authors should further justify why this energy-only representation is sufficient for all musical genres, especially those with unique timbral characteristics.

---

> ### Author Rebuttal · Authors · 2026-03-30
>
> We would like to thank the reviewer for taking the time to evaluate our work and for the positive comments on its motivation and creativity. Below are our responses to the comments in Weaknesses and Questions.
>
> > W1. A more systematic, data-driven approach for threshold selection—or a discussion on its stability—would strengthen the practical deployment of the detector.
>
> In our current implementation, the decision threshold is not a carefully tuned constant tied to a specific test set. Instead, it can be selected in a data-driven way based on the likelihood distribution of real training samples. A natural choice is a quantile-based thresholding scheme, where the threshold is determined by a target acceptance rate for real music. This strategy is simple, reproducible, and well-aligned with the one-class nature of our method, as the model is designed solely to characterize the distribution of real music. We evaluate the detector across thresholds around the operating point of -20 used in the paper. As shown below, accuracy remains above 98% over a wide range, indicating limited sensitivity to threshold variation and no reliance on a finely tuned operating point.
> |Threshold|-10|-15|-20|-25|-30|
> |-|-|-|-|-|-|
> |Acc. (%)|98.1|100|99.9|99.8|99.7|
>
> > W2. The authors should further justify why this energy-only representation is sufficient for all musical genres, especially those with unique timbral characteristics.
>
> We agree that phase information and deeper temporal dependencies may provide complementary cues. However, for our likelihood-based method, **the key is not to model all signal details, but to capture the dominant statistical discrepancies** between real and generated music. Our core argument is that, although timbral details exhibit significant diversity, real music follows stable statistical regularities in its frequency-domain energy distribution. These regularities, which reflect underlying physical and perceptual principles, are well suited to be captured by normalizing flows.
>
> The energy spectrogram encodes not only frequency distribution, but also harmonic structure, cross-band dependencies, spectral envelope patterns, high-frequency details, and temporal continuity. While these characteristics vary across styles and instruments, they remain governed by shared structural constraints in real-world audio signals. Accordingly, **MusicDET focuses on modeling the overall distribution of real music rather than superficial cues tied to specific instruments or styles**. In contrast, generated music often exhibits systematic deviations in spectral energy distribution and cross-frequency dependencies, which can be effectively captured in the learned likelihood space. Our cross-genre and cross-instrument experiments (please refer to Reviewer 56G8 W1Q1) further support this point. We also agree that incorporating phase information or longer temporal modeling may further improve robustness, which we view as a promising direction for future work.
>
> > Q1. To better validate the "zero-shot" and "generator-agnostic" claims of MusicDET, could you provide detection results (EER) for these newer models?
>
> The advanced models YuE[1] and ACE-Step[2] differ in their generation paradigms and music characteristics, providing a meaningful test of our method’s generalization ability. Specifically, we construct a diverse test set spanning multiple musical genres (e.g., pop, rock, hip-hop), including both vocal (male/female) and instrumental samples. For each prompt, the generation models produce 10-second music clips. Notably, MusicDET is trained only on real music from SONICS, which is collected from YouTube, and does not include any samples from YuE[1] or ACE-Step[2], thereby ensuring a strict zero-shot setting. By contrast, all competing methods are trained on the same real-music set together with AI-generated music from Suno v3.5. For testing, the real music is drawn from MusicCaps, while the AI-generated music is produced by YuE[1] and ACE-Step[2], respectively. The resulting EER (↓) is summarized below:
> |Method|YuE (%)|ACE-Step (%)|
> |-|-|-|
> |AASIST|23.3|23.8|
> |MERT-AASIST|24.7|40.7|
> |W2V2-AASIST|37.3|30.6|
> |WPT-W2V2-AASIST|42.9|17.5|
> |Spec-ViT|43.3|33.3|
> |Spec-ConvNeXt|39.3|43.5|
> |Spec-TTTra-α|47.3|20.9|
> |Spec-TTTra-β|34.6|19.8|
> |Spec-TTTra-γ|40.2|20.4|
> |MusicDET|6.6|0.6|
>
> We observe that when both the real and AI-generated music come from previously unseen sources, these classification-based methods generally struggle to separate the two reliably. In contrast, MusicDET achieves 6.6% EER on YuE[1] and 0.6% EER on ACE-Step[2] under a zero-shot setting, demonstrating strong robustness and excellent cross-generator generalization. The relevant code, prompts, and metadata will be released for reproducibility.
>
> [1] YuE: Scaling Open Foundation Models for Long-Form Music Generation
>
> [2] ACE-Step: A Step Towards Music Generation Foundation Model

---

> > ### Author Rebuttal · Reviewer_qtWt · 2026-04-05
> >
> > Thanks for the response. Most of my concerns have been addressed.

---

> > > ### Author Response · Authors · 2026-04-06
> > >
> > > We are very pleased to hear that our response has addressed the reviewer’s concerns.
> > > We sincerely thank the reviewer for their time, effort, and valuable feedback.

---

### Decision · Program_Chairs · 2026-04-30

**Decision:**

Accept (regular)

**Comment:**

The paper presents MusicDET, a likelihood-based framework for zero-shot AI-generated music detection that models the distribution of real music, with strong empirical results and good generalization to unseen generators.

All reviewers agree the approach is well-motivated, clearly presented, and tackles an important problem, with novel approach through probabilistic formulation. At the same time, reviewers raised a few concerns about the motivation of the proposed approach, some aspects of the evaluation (e.g., near-zero EER results, lack of latest generation models, and limited analysis of vocal vs. instrumental music), and missing details for reproducibility. The authors addressed key concerns during the rebuttal, and the paper represents a solid contribution with practical relevance. I highly recommend the authors to include these details in the final manuscript.

Finally, I recommend acceptance.